# Social Support between Diabetes Patients and Non-Diabetes Persons in Yangon, Myanmar: A Study Applying ENRICHD Social Support Instrument

**DOI:** 10.3390/ijerph18147302

**Published:** 2021-07-08

**Authors:** Ei Thinzar Khin, Myo Nyein Aung, Satomi Ueno, Ishtiaq Ahmad, Tint Swe Latt, Saiyud Moolphate, Motoyuki Yuasa

**Affiliations:** 1Department of Public Health, Graduate School of Medicine, Juntendo University, Tokyo 113-8421, Japan; k-ei@juntendo.ac.jp (E.T.K.); ueno@sjc-nagano.ac.jp (S.U.); ahmad@juntendo.ac.jp (I.A.); moyuasa@juntendo.ac.jp (M.Y.); 2Advanced Research Institute for Health Sciences, Juntendo University, Tokyo 113-8421, Japan; 3Faculty of International Liberal Arts, Juntendo University, Tokyo 113-8421, Japan; 4Faculty of Nursing, Seisen Jogakuin College, Nagano 381-0085, Japan; 5Myanmar Diabetes Association (MMDA), Yangon 11211, Myanmar; proftsl@gmail.com; 6Department of Public Health, Faculty of Science and Technology, Chiang Mai Rajabhat University, Chiangmai 50300, Thailand; saiyudmoolphate@gmail.com

**Keywords:** non-communicable diseases, NCD, family, household, public health, global, community, type2 diabetes, Myanmar

## Abstract

Diabetes patients, due to the chorionic nature of the disease, need complex and long-term care for control and prevention of complications. The patients themselves find it difficult to adopt appropriate disease management after diagnosis and they need social support from family, friends, and their environment, especially in lower- and middle-income countries where medical service is limited, and they need self-care of disease and lifestyle modification. In Myanmar, however, the study for social support among diabetes patients is still limited. Therefore, we conducted a case-control study to investigate the social support among diabetes patients and the association between socioeconomic factors in Yangon, which has the highest prevalence of diabetes in Myanmar. Social support between diabetes patients who came to diabetes special clinics and non-diabetes community control was assessed by applying transculturally translated ENRICHD Social Support Instrument (ESSI). Among the diabetes patients’ group, more than 70% had high perceived social support, specifically higher level of informational and emotional social support. Robust multiple regression models revealed significant positive associations between total social support and independent variables: *p* value < 0.001 for monthly household income and being married, and *p* value < 0.05 for household number and frequency of having meals together with family. These findings suggest that perceived social support among patients with diabetes may be mainly affected by the patients’ family conditions, such as household income and living with a spouse, in Myanmar culture.

## 1. Introduction

At present, diabetes is known as one of the major public health concerns in the third millennium and is the fifth main mortality cause in the world [1,2]. In 2017, it was assessed that there are 451 million (age 18–99 years) individuals with diabetes worldwide and nearly a half (49.7%) living with diabetes are undiagnosed [3]. Furthermore, 75% of those with diabetes were living in low- and middle-income countries [4]. At present, Myanmar, which is also a low-income Southeast Asian country with a population of 53.9 million, is experiencing a surge in diagnosis of diabetes and other non-communicable diseases [5]. According to the National Survey of Diabetes and its Risk Factors conducted in 2014, the prevalence of diabetes in Myanmar was 10.5% for the adult population (age 25–74 years) and the estimated number of adults with diabetes was 2.5 million, relatively higher than that reported in many Association of Southeast Asian Nations countries [6].

Although diabetes is an incurable disease, it can be treated and controlled to prevent its complications, and some people may go into remission by lifestyle changes. However, because of the severe and persistent nature of diabetes, the difficulty of its treatment, and the many everyday self-care decisions that diabetes necessitates, sticking to a predetermined care regimen is usually insufficient over the course of a person’s diabetes existence [7]. After the diagnosis with diabetes, in order to adapt lifestyle changes and achieve effective disease management, the social world of the patient becomes an important factor to support and buffer these impacts which is not sufficient with only treatment from healthcare providers. In the UK, people with diabetes typically have only a few clinic visits each year, with a total of less than 2 h of formal diabetes care. They manage the overwhelming majority of their diabetes-related needs on their own [8]. Recently, many countries developed and adapted self-management guidelines for diabetes, but the patients themselves alone find it difficult to carry out those guidelines. Therefore, the support from social relations with friends, peers, and family can help daily lifestyle adaptation and diabetes care, especially in low income countries where formal medical care is not sufficient [9].

A higher level of social support has also been linked to improved diabetes self-management in many previous trials [10,11,12,13]. The term “social assistance” refers to a resource exchange between at least two individuals in which one of them believes it is meant to improve the recipient’s well-being [14]. Social support includes all instrumental aid, emotional support, encouragement, and informative advice available through relationships and environment, e.g., family members, healthcare providers, and peers or friends. Social support can also be specified in terms of received and perceived social support. Perceived support refers to one’s potential access to social support, whereas received support refers to the reported receipt of support resources, usually during a specific time frame [15]. According to Sathish, T. et.al, Kerala Diabetes Prevention Program conducted in India has demonstrated that implementing a low-cost peer-support lifestyle intervention in community settings for individuals at high risk of developing diabetes was cost-effective in a low- and middle-income setting in terms of number of diabetes cases prevented, decreased number of out-patient visits, and higher QALY, Quality Adjusted Life Year [16]. In addition, the International Diabetes Federation confirms that poor social support is a predictor of poor adherence to the prescribed therapy.

Meanwhile, although social support has been studied as an independent and moderating factor during the last few years, a relatively few studies were conducted in a comparison to non-diabetes, to find out an association between the social support of family and environment to the patients’ diabetes status. The determinants of social relationships, as well as their consequences, are crucial to the theoretical and causal status of social relationships in relation to health. Understanding the pathways through which other variables influence social support and how they interact with each other on diabetes outcome and management among type 2 diabetes patients will aid in developing more effective interventions in the family or community for this growing population and reducing the burden of healthcare service for NCD in the country.

In Myanmar, there are previous studies investigating the association between social support and infectious diseases such as HIV [17] and malaria [18], but regarding social support of people with diabetes and NCDs, such a report has still been limited. Evidence on social support is even more important for the setting in a developing country like Myanmar because of its potential to constitute a low-cost intervention to promote health, in primary care, and self-management [19]. Therefore, we performed a case-control study (*n* = 300) to evaluate the standard of social support and the association between social support and other demographic factors and health knowledge after diagnosis of diabetes in a Myanmar population.

## 2. Materials and Methods

### 2.1. Study Design

This study was a case-control study. A total of 300 participants (150 cases) and (150 controls) were recruited in the data collection process. The study population consisted of people aged from 25 to 74 years of both sexes residing in Yangon, Myanmar. The study excluded the following people: seriously ill people (e.g., kidney diseases, hospitalized patients), long-term modification of diet due to any reason (e.g., meditation or taking Sabbath), pregnant women, institutionalized individuals, and temporary residents (those living in Yangon region for less than 6 months). Our hypothesis of the study was that there will be a difference in social support between cases (diabetes patients) and control (people without diabetes).

#### 2.1.1. Selection Criteria for Case Group

Patients between 25–74 years of age who were recently diagnosed with type 2 diabetes mellitus within 6 months were recruited as the case at diabetes special clinics of 2 hospitals (xxx and yyy) in Yangon. The diagnosis criteria were according to WHO definition and diagnosis of diabetes mellitus and intermittent hyperglycemia, i.e., fasting plasma glucose ≥126 mg/dL and/or random blood glucose (RBG) ≥200 mg/dL, with or without osmotic symptoms of diabetes (polyuria, polydipsia, thirst, body weight loss) [20].

#### 2.1.2. Selection Criteria for Control Group

A community control with the same age as the case, 25–74 years, was recruited after a test for screening diabetes was negative. Control group data were collected randomly from xxx township, Yangon. The community engagement process started through the community leader informing about the study and getting informed consent in advance before data collection. The RBG of each participant was checked in the laboratory, and if the RBG < 140 mg/dL, we recruited them as a control subject. If the participant was newly diagnosed with diabetes from our screening process, we properly referred him/her to the diabetes clinic or physician for further treatment.

### 2.2. Measurements

#### 2.2.1. Assessment of Social Support

We used the ENRICHD Social Support Inventory (ESSI) to assess social support among participants. The full-length ESSI is a 7-item self-report survey [21], that was developed to screen post-myocardial infarction patients for low levels of perceived support as part of the inclusion criterion for the Enhancing Recovery in Coronary Heart Disease (ENRICHD) trial [22]. The standardized coefficient alpha for the ESSI is 0.87 [21]. It measured the four defining attributes of social support: the emotional, instrumental, informational, and appraisal by using the questions, for example “Is there someone available to you whom you can count on to listen to you when you need to talk?”, “Is there someone available to give you good advice about a problem?”. The response options of each item ranged from 1 (none of the time) to 5 (all of the time) except the question of “Are you currently married or living with a partner?” which is a Yes/No type of question. According to the criteria of the ENRICHD protocol (Version 7.0), low perceived social support was defined as a total score of ≤18 and a score ≤3 in terms of at least 2 items, excluding items 4 and 7 [23]. For this study, we studied both perceived social support (low or high) from total scoring and also each item of social support as continuous data. The internal consistency (Cronbach’s alpha) of the Myanmar version of the 7-item ESSI was 0.854 from the pilot study in which we tested the feasibility, reliability, and validity of the proposed study design.

#### 2.2.2. Assessment of Dietary Habits

Dietary habits of the participants were assessed by self-administered questionnaires. The frequency of meals with family was measured using the question “Over the past week, how often did you have meals with your family?”. Response options were “never/very rarely”, “1–2 times/week”, “3–4 times/week”, “5–6 times/week”, or “everyday/week”. The frequency of eating out was assessed by using the question “Over the past week, how often did you eat out?” with the same response options. We analyzed these two questions into categorical data as “less than 3 times/week”, “3–6 times/week”, and “everyday/week”. Meal frequency per day, breakfast eating behaviors, and intake of fast food and takeout food were also assessed.

#### 2.2.3. Other Covariates Assessments

The study questionnaire contained demographic and socioeconomic factors including age, sex, ethnicity, education level, occupation, marital status, number of household members, individual income, and household income. The gender was coded as male and female categories. The ethnicity of the participants was assessed by 8 categories including Burmese and other 7 major ethnic groups in Myanmar, then recategorized into Burmese and other ethnicities. Education was categorized as elementary school graduate or less, middle school graduate, high school graduate, college graduate, and university graduate or higher. The main work status over the past 12 months was assessed by various categories including government employee, self-employed, skilled workers, and unemployed or elderly, etc. Then, the income status was categorized as lower normal, normal, and upper normal according to the local setting and income calculation of Central Statistical Organization, Myanmar. Health-related behaviors were investigated for smoking behavior and alcohol consumption. The questions on smoking and alcohol drinking were based on the WHO STEP-wise approach to non-communicable disease risk-factor surveillance. For medical history, the participants were asked for a history of hypertension, complications of diabetes, and family history of diabetes. Anthropometric measurements including body mass index (BMI), central obesity, and blood pressure of the participants were measured. Central obesity was calculated by sex-specific waist–hip ratio. Waist–hip ratios ≥0.9 in males and ≥0.85 in females were regarded as central obesity by WHO cut-off points. Blood pressure was measured by the oscillometric method by using OMRON digital automatic blood pressure monitor HEM-907 and hypertension was diagnosed if the systolic blood pressure readings were ≥140 mmHg and/or the diastolic blood pressure readings ≥90 mmHg upon two repeated measurements. Blood pressure measurements were categorized into two groups, normal BP (<140/90 mmHg) and high BP (≥140/90 mmHg) [24].

### 2.3. Data Collection

Before the study in Myanmar, all the development of instruments and the pilot test with 30 participants was conducted among Myanmar people who were living in Tokyo, Japan. According to the WHO process of translation and adaptation of instruments [25,26], all the questionnaires were translated to Myanmar languages and back translated into English and retranslated into Myanmar by independent language experts and edited by researchers. The study was conducted from February 2019 to March 2019 in Yangon. During the time of the interview, a trained research assistant asked the participants with a structured questionnaire. After the interview, health assessments for height and weight, BMI, waist and hip ratio, body fat percentage, and blood pressure were conducted. For the control group, a random blood glucose test was checked by taking a venous blood sample and sent to a standard laboratory within 24 h for diabetes screen purpose.

### 2.4. Statistical Analysis

Data entry followed double entry and validation. Continuous variables were expressed as a mean with standard deviation, and categorical variables as a percentage. Some continuous variables were made into dichotomous categorical variables by using suitable standard cut-offs. The t-test and the chi-square test were used to compare continuous and categorical differences between the case and control group, respectively (Table 1 and Table 2). Multivariate analysis of variance (MANOVA) was applied to compare six ESSI items in a single analysis (Table 3). Robust multivariable regression analyses were conducted to investigate the association between social support of people with diabetes and other demographic and socioeconomic factors. The dependent variable “social support” was considered as the continuous variable. Three models of robust multivariate regression analyses—model 1, model 2, and model 3 (Table 4)—were applied to test the association between the dependent and independent variables with adjustment for various confounders. Covariates in the model 1 were case/control, age, sex, family income, and educational attainment; in model 2 were variables in the model 1 plus marital status, household number, and mealtime with family; in the model 3 were variables in the model 2 plus alcohol consumption and family history of diabetes. *p*-value <0.05 was considered as statistically significant with a 95% confidence interval. All statistical analyses were performed with the STATA version 17 SE (Stata Corporation, College Station, TX, USA).

## 3. Results

### 3.1. Basic Characteristics of the Participants

We compared the characteristic of 150 newly diagnosed diabetes patients and 150 community residents without diabetes. Diabetes patients were older, more uneducated, more unemployed, and had less household income than the non-diabetes participants (Table 1). Those differences are statistically significant. More than 80% of the ethnicity of both groups was Burmese. General education status in control group was higher than in the case group (Table 1) since 33.3% of diabetes patients had education up to elementary school and only 22.0% of the diabetes patients had a college education, although 8.7% of control group had education up to elementary school and 60.0% of those went to college or graduate school. Nearly half of diabetes patients (54.0%) were elderly or dependent and most of control group (66.7%) were currently employed. Household number in both groups had no significant difference, but the household income is significantly greater than in the control group (Table 1). Family history of diabetes was more common among the diabetes patients, whereas 12.0% of non-diabetes participants did not know whether their families had a history of diabetes.

### 3.2. Health Assessment of the Participants

For the dietary habits, 67.3% of the total participants had an eating out frequency less than three times/week. The case group (44.0%) had more times on having meals with family than the control group (24.0%) (Table 2). The mean BMI of the case and control is 26.78 kg/m^2^ (SD = 4.94) and 24.94 kg/m^2^ (SD = 4.59), receptively. A percentage of 53.3% of the case group had central obesity, and only 31.3% of the control group had central obesity. The proportion of hypertension was 24.0% and 17.0% among cases and controls. The smoking rate is significantly lower among the diabetes patients than the non-diabetes participants and likewise is the alcohol consumption (Table 2).

### 3.3. Status of Social Support in Case and Control Group

Each item of ENRICHD social support instrument from item 1 to 6 was described as continuous data and the mean with standard deviation was calculated. Item 7, which is living with spouse or other relatives, was dichotomous data and revealed with percentage. From items 1 to 6, item 2 (someone available to give good advice about a problem) and item 6 (someone in whom can trust and confide in) had a significant difference between case and control group The mean score of item 2 in the case group, 3.95 (SD = 1.19), is higher than that in control group, 3.77 (SD = 1.06) (*p* < 0.05, MANOVA), while the mean score of item 6 in case group, 4.21 (SD = 1.1.2), was higher than that in control group 3.8 (SD = 1.14) (*p* < 0.01, MANOVA). For the item 7, 91.3% of diabetes patients lived together with their spouse or other relatives while 77.0% of non-diabetes persons lived with their spouse or other relatives (*p*-value <0.001, chi-squared) (Table 3). The total social support score of case group with the mean value of 23.67 (SD = 5.33) was higher than that of control group with the mean value of 22.85 (SD = 5.75) (*p*-value = 0.16). For the level of perceived social support, 28.67% of the case group and 37.33% of the control group had low perceived social support. (Table 3)

The score of social support was significantly associated with presence of diabetes in model 1 of robust multiple regression analysis (Table 4). In models 2 and 3, the household income and married life of the participants have a statistically significant positive association with total social support (*p*-value < 0.001). In models 2 and 3, the participant’s household number and frequent mealtime with family showed a positive significant association with social support score (*p*-value <0.05).

## 4. Discussion

We have performed an observational study in the Yangon population, Myanmar with a total of 300 participants to measure the social support of diabetes patients and non-diabetes community people aged between 25–74 years. The prevalence of diabetes in the Yangon region was 18.2% [6] and was higher than twice the global (9.3%) and SEAR, South East Asia Regions (8.8%) estimates in 2019 [3]. Since there was an alarming condition of diabetes in Yangon, there were many articles of research in this region that have been studied for diabetes, for example, risk factors for non-communicable diseases [27] and factors influencing adherence to therapeutic regimens among people with type 2 diabetes [28]. However, there has been no previous study of social support of diabetes patients in Myanmar; thus, we conducted this study to investigate social support among diabetes patients, which is also an influential factor for type 2 diabetes management. In addition, we assessed what types of social support made the difference and what factors are influencing the perceived social support.

According to the Myanmar Population and Housing Census 2014, there was a total population of over 51 million people in Myanmar and its annual growth rate of population was estimated to 0.89% between 2003 and 2014 [29]. Myanmar has over 135 different tribes, and each ethnic group has its own history and traditions of living styles. Among 14 administrative divisions, Yangon division was the most populated area with nearly 7.4 million (14.3% of the country). Moreover, Yangon, the country’s largest city, is home to various businesses and a unique blend of culture and customs. Due to its dense population, people in the communities are usually living together with their family, relatives, or friends, and most are an extended type of household unit. All participants in our study were community residents living in Yangon, and the demographic finding showed that there were an average of five members in a household. The finding may reflect the current living society of metropolitan Yangon (Table 1).

During recent decades of years, population growth and urbanization have had some major influences on the socioeconomic condition of each country and the lifestyle of its people. Although urbanization leads to better access of education, career development, and healthcare services, it also has adverse effects such as pollution, unhealthy dietary habits, more sedentary lifestyle, and social deprivation [30]. According to a previous study by Aung et al., the prevalence of type 2 diabetes mellitus in urban areas of the Yangon Region was significantly higher in urban than in rural areas, respectively, 12.1% and 7.1% [31]. Beyond the lifestyle risk factors such as smoking, insufficient physical activity, and less consumption of vegetables, the authors suggested that urban stress might be one of the reasons for higher diabetes prevalence of urban residents. On a daily basis, many of urban residents face overcrowding, unemployment, poor housing, poverty, competition, and cultural dislocation, which contributes to stress conditions such as anxiety and depression [32]. The Japan Gerontological Evaluation Study (JAGES) recently conducted in Myanmar 2018 reported that elderly people who lived in Yangon (representative of urban) rarely met friends, scored higher for instrumental activities of daily living, higher for being housebound, and lower for social role than those who lived in Bago (representative of rural) [33]. Therefore, it becomes necessary to identify the factors embedded in local social and cultural context for the intervention of better health outcomes.

In this study, we studied to find out the basic socioeconomic factors of recently diagnosed diabetes patients such as employment condition, household income, and social status in most urbanized regions of Myanmar, Yangon (Table 1). Over 80% of the participants were Burmese people, which is the major ethnicity of the country. Among diabetes patients, only 22% had a general education of college or university, in contrast to non-diabetes persons, 60% of whom had a higher education level. A percentage of 44.57% of diabetes patients were employed, and 54% were unemployed or elderly. Although there was no significant difference for household number between the case and control groups, most of the diabetes patients (64%) had a normal household income and most of the controls (59.33%) had an upper normal household income. It may be a favorable condition for social support from family.

Socio-demographic characteristics showed that there is a convergence of several social determinants of health being poor among the diabetes group. Relatively, they attended fewer school years than the non-diabetes persons (Table 2). Almost half of them were unemployed. Given the poorer determinants of health among the diabetes patients (Table 1), metabolic indexes in Table 2 are suggesting inequalities in health outcomes. They are more obese, having a higher mean BMI and bigger waist than the non-diabetes group. Their blood pressure was higher than that of non-diabetes participants (Table 3). According to a recent report, income inequality and poverty gap is serious in Myanmar. It is one of countries with the highest poverty gap, with a Gini index of 30.1 according to a World Bank report in 2017 [25]. Our finding addressed an important health inequality to address the combination of “the poorer, the less educated, the more obese, and diabetes” to prevent diabetes in Myanmar.

In contrast, there was a lower smoking rate and alcohol consumption rate among the diabetes patients. Perhaps these might be the impact of diabetes-related health education. Importantly, there were more females in the diabetes group, highlighting gender inequality. However, we found that social support was not different between two genders.

According to the first Oxford International Diabetes Summit (2002), it has recommended psycho-social aspects of diabetes to be included in an individual country’s national guidelines [34]. In a previous study, diabetes patients with psychological distress were found to have higher cardiovascular disease, CVD morbidity, and mortality rates compared to those without psychological distress over 5.4 years of follow-up [35]. For the life-long control of diabetes and prevention of its complications, patients may need not only therapeutic measurements and lifestyle modification, but also psychosocial intervention. A 2018 study that studied the association between diabetes distress and glycemic control further confirmed that glycemic control could be improved with effective approaches in reducing diabetes distress and enhancing social support from family and friends [36]. In our study, based on ESSI calculation, more than half of the diabetes patients and non-diabetes participants had high perceived social support, with 71.33% and 62.67%, respectively (Table 3). This shows that most of the participants have high perceived social support.

Multivariate analysis comparing all ESSI constructs holistically between diabetes and non-diabetes revealed a statistical significance. A significant difference between the two groups was in the informational support for diabetes patients, in which they had someone available to give good advice about a problem (ESSI question no. 2) and someone in whom they trust and confide (ESSI question no.6) for almost all of the time (Table 3). This result agrees with an earlier qualitative study in Malaysia where they found that social influences from family members, friends and peers, and health care providers played a main source of information that supported patients’ decision-making in choosing the type of treatment on diagnosis [37]. It may assist the patients’ health-seeking behaviors and delivery of health information to the patients. On the other hand, the patient’s level of confidence in their social network often played a role in their willingness to consider suggestions. Patients will be more likely to believe, consider, and follow through on advice and suggestions if they have confidence in their human relationships [38]. Therefore, our findings can be applied in the daily life and social environment of diabetes patients to give advice and knowledge sharing for disease management and psychosocial support from their trustful person/s such as family, friends, or peers.

Individuals perceived to acquire three primary types of social support in the context of interpersonal relationships: emotional, informational, and tangible [19]. In this study, two types of social supports were significantly higher for diabetes patients than the non-diabetes participants. These were the informational type of social support (Q2 Table 3) and emotional support (Q6 Table 3). The origin of such social support can be either from a newly acquired network of peer-diabetes or from the family, friends, and existing social networks belonging to individuals. With a new diagnosis, people may meet peers who have the same disease. It is a very useful form of social support, especially for sharing experiences, lifestyle adaptation, and informational help. Existing literature said that friendship serves as a behavioral vaccine primarily through its social support effect [19]. Moreover, sympathy of family and friends may result in positive social support, especially to understand the feelings of the patients. Multivariable analysis identified factors influencing the social support: household income, marital status, and mealtime with family (Table 4). Diabetes status, age, and sex were adjusted in the analysis.

We found that over 90% of diabetes patients in our study were married or currently living with a partner, a relative, or a friend (Table 3). In addition, marital status, household income, and mealtime with family were found to influence social support positively (Table 4). In a previous study, it had been proved that marital status per se was an important predictor of risk factors for cardiovascular diseases among unmarried men but not women. Unmarried men or men living alone had lower HDL cholesterol level and a higher systolic blood pressure reading than married men [39]. Differences in dietary consumption between married and unmarried men may be such that married men pick up on some of their wives’ healthier eating habits, for example, intake of polyunsaturated fat and dietary fibers [40].

Eating meals together with family is advantageous to our health [41]. The benefit of eating meals together with families were reported in terms of nutritional values. A study in the US. reported that people who have more frequent dinners with family tended to eat a healthy diet and thus family dinners may be an appropriate intervention for a healthy lifestyle [41]. Another study in Japan reported that eating meals alone was associated with frailty among older people [42]. In our study, we have an important finding that mealtime with family also had a positive influence on the social support of the diabetes patients (Model 2, Table 4). Our finding consolidated the evidence that family meals are a source of psychosocial benefit for better health. Furthermore, according to Epple C. et.al., active family nutritional support had a positive influence on the clinical measure of metabolic control and better diabetes outcomes for HbA1c level, triglyceride, and total cholesterol level [43]. Myanmar families usually have meals together at home in both urban and rural areas. Despite increasing trends of urban lifestyle, only 10.67% of all participants went for eating out every day, and 67.33% did fewer than three times/week (Table 2). In a previous study, the perceived social support received from family specific to the diet was significantly associated with glucose control among Korean immigrants with diabetes [44]. Therefore, our finding is expected to suggest a possible delivery of family-based health promotion lifestyle intervention to prevent diabetes in Myanmar and countries sharing a similar culture in the future.

This study was the first study of social support for diabetes patients in the urban area of Yangon and found out important associations between socioeconomic factors and family status. Strengths of the study were eligibility criteria to select (1) the case and control group, (2) community control sustaining the distribution of parameters in target population, (3) measurement using ESSI scale, which investigated the recent event and minimized the possible recall bias, and (4) multivariable analysis adjusting several covariates associated with social support. Since we followed steps of transcultural translation to adopt ESSI questionaries for Myanmar culture [25] and its validity and reliability were checked by researchers, this instrument can be used in the further study of social support study in Myanmar.

This study may have limitations. As is the nature of a cross-sectional study, the association we reported in this study has a lack of temporality and is required to be judged with biological and logical plausibility [45]. In the case group of diabetes patients, recent diagnosis with diabetes and health education that was recently acquired from a diabetes clinic may influence social status and health behavior. Longitudinal studies examining the need for social support of people with diabetes and the control group and social support intervention studies are recommended for future research.

The results of the current research are hoped to benefit public health and psychosocial care services of not only diabetes but also of other non-communicable diseases. The descriptive and analytical findings of the study are supposed to provide a practical inspiration for developing evidence-based intervention programs in Myanmar, and in other Southeast Asian countries, that will conserve and promote the social supports of diabetes patients, thereby reducing the psychosocial distress of the patients and improving health-seeking behaviors of the patients.

## 5. Conclusions

In our current study, the perceived social support status of diabetes patients was higher, and they received more informational support, including advice and confidences, than people with no diabetes. Since the high level of social status of the patients has a positive association with marital status, family income, and mealtime with family, future research should focus on the intervention on a family or household level to modify a healthy lifestyle, to share helpful advice and information, and to reduce the impact of psychosocial stress among the patients.

## Figures and Tables

**Table 1 ijerph-18-07302-t001:** Basic characteristics of the participants.

	Case (*n* = 150)	Control (*n* = 150)	Total (*n* = 300)	*p*-Value
Sex, *n* (%)				
Male	47 (31.3)	67 (44.7)	114 (38.0)	0.02
Female	103 (68.7)	83 (55.3)	186 (62.0)	
Age (mean ± SD)	55 (10.9)	43 (14.8)	49 (14.2)	<0.001
Ethnicity, *n* (%)				
Burmese	125 (83.3)	131 (87.3)	256 (85.3)	0.33
Other	25 (16.7)	19 (12.7)	44 (14.7)	
Education status, *n* (%)				
Up to elementary school	50 (33.3)	13 (8.7)	63 (21.0)	<0.001
Up to high school	67 (44.7)	47 (31.3)	114 (38.0)	
College and graduate	33 (22.0)	90 (60.0)	123 (41.0)	
Employment, *n* (%)				
Currently employed	67 (44.7)	100 (66.7)	167 (55.7)	<0.001
Unemployed or elderly	81 (54.0)	41 (27.3)	122 (40.7)	
Other	2 (1.3)	9 (6.0)	11 (3.6)	
Household number (mean ± SD)	4.49 (2.5)	4.32 (1.8)	4.40 (2.2)	0.56
Household income, *n* (%)				
Lower normal	11 (7.3)	4 (2.7)	15 (5.0)	<0.001
Normal	96 (64.0)	57 (38.0)	153 (51.0)	
Upper normal	43 (28.7)	89 (59.3)	132 (44.0)	
Family history of diabetes, *n* (%)				
Yes	89 (59.3)	69 (46.0)	158 (52.7)	0.01
No	56 (37.3)	63 (42.0)	119 (39.7)	
Not known	5 (3.3)	18 (12.0)	23 (7.6)	

Abbreviation: SD, Standard Deviation. Note: Student’s t-test or Mann–Whitney U test was used to compare continuous variables, and chi-square test was used to compare categorical variables between case and control. *p* value < 0.05 statistically significant, *p* value < 0.001 statistically strongly significant.

**Table 2 ijerph-18-07302-t002:** Health assessments of the participants.

	Case (*n* = 150)	Control (*n* = 150)	Total (*n* = 300)	*p*-Value
Mealtime with family, *n* (%)				
Less than 3 times/week	55 (36.7)	65 (43.3)	120 (40.0)	<0.001
3–6 times/week	29 (19.3)	49 (32.7)	78 (26.0)	
Every day	66 (44.0)	36 (24.0)	102 (34.0)	
Eating out, *n* (%)				
Less than 3 times/week	104 (69.3)	98 (65.3)	202 (67.3)	0.76
3–6 times/week	31 (20.7)	35 (23.3)	66 (22.0)	
Every day	15 (10.0)	17 (11.3)	32 (10.7)	
BMI (kg/m^2^), (mean ± SD)	26.78 (4.94)	24.94 (4.59)	25.86 (4.86)	0.01
Central obesity, *n* (%)				
Yes	80 (53.3)	47 (31.3)	127 (57.7)	<0.001
No	70 (46.7)	103 (68.7)	173 (42.3)	
Blood pressure (mmHg), *n* (%)				
Normal BP (<140/90)	114 (76.0)	124 (82.0)	238 (79.3)	0.15
High BP (≥140/90)	36 (24.0)	26 (17.0)	62 (20.7)	
Smoking tobacco, *n* (%)				
Never smoked	125 (83.3)	121 (80.7)	246 (82.0)	0.002
Former smoker	21 (14.0)	11 (7.3)	32 (10.7)	
Current smoker	4 (2.7)	18 (12.0)	22 (7.3)	
Alcohol, *n* (%)				
Never drink	130 (86.7)	103 (68.7)	233 (77.7)	<0.001
Quit drinking	14 (9.3)	7 (4.7)	21 (7.0)	
Still drinking	6 (4.0)	40 (26.6)	46 (15.3)	

Abbreviation: SD, Standard Deviation. Note: Simple t-test or Mann–Whitney U test was used to compare continuous variables, and chi-square test was used to compare categorical variables between case and control. *p* value < 0.05 statistically significant, *p* value < 0.001 statistically strongly significant.

**Table 3 ijerph-18-07302-t003:** Social support between case and control.

ESSI Questions			Case (*n* = 150)	Control (*n* = 150)	*p*-Value
Q 1. Is there someone available to you whom you can count on to listen to you when you need to talk?		(mean ± SD)	3.84 (1.18)	3.67 (1.18)	0.16
Q 2. Is there someone available to give you good advice about a problem?		(mean ± SD)	3.95 (1.19)	3.77 (1.06)	<0.05
Q 3. Is there someone available to you who shows you love and affection?		(mean ± SD)	4.14 (1.03)	3.99 (1.18)	0.39
Q 4. Is there someone available to help you with daily chores?		(mean ± SD)	2.7 (1.63)	3.03 (1.55)	0.08
Q 5. Can you count on anyone to provide you with emotional support (talking over problem or helping you make a difficult decision)?		(mean ± SD)	3.92 (1.13)	3.81 (1.05)	0.16
Q 6. Do you have as much contact as you would like with someone you feel close to, someone in whom you can trust and confide?		(mean ± SD)	4.21 (1.1.2)	3.8 (1.14)	<0.001
Q 7. Are you currently married or living with a partner?	Yes	(*n*%)	137 (91.3)	114 (77.0)	<0.001
	No		13 (8.7)	34 (23.0)	
Total social support scores (Q 1–7)		(mean ± SD)	23.67 (5.33)	22.85 (5.75)	0.16
Perceived social support	High	(*n*%)	107 (71.3)	94 (62.7)	0.11
	Low		43 (28.7)	56 (37.3)	

Note: Test used in analysis was MANOVA test. *p* value < 0.05 statistically significant, *p* value < 0.001 statistically strongly significant.

**Table 4 ijerph-18-07302-t004:** Robust multivariable regression analyses showing the relation between diabetes and social support defined by the potential confounders.

	Univariate		Model 1		Model 2		Model 3	
	Coef. (95% CI)	*p*-Value	Coef. (95% CI)	*p*-Value	Coef. (95% CI)	*p*-Value	Coef. (95% CI)	*p*-Value
Case-control ^#^	0.79 (−0.40 to 1.99)	0.19	1.35 (0.03 to 2.67)	0.05	0.82 (−0.478 to 2.12)	0.22	0.75 (−0.57 to 2.07)	0.26
Age	0.02 (−0.02 to 0.07)	0.27	0.04 (−0.00 to 0.09)	0.08	0.03 (−0.02 to 0.08)	0.18	0.03 (−0.01 to 0.08)	0.14
Sex	0.67 (−0.54 to 1.88)	0.28	0.59 (−0.62 to 1.80)	0.34	0.67 (−0.51 to 1.86)	0.27	0.79 (−0.69 to 2.28)	0.29
Household income	1.67 (0.97 to 2.37)	<0.001	1.98 (1.24 to 2.71)	<0.001	1.74 (1.02 to 2.45)	<0.001	1.70 (0.98 to 2.43)	<0.001
Education	0.46 (−0.33 to 1.25)	0.25	0.66 (−0.23 to 1.55)	0.15	0.83 (−0.03 to 1.68)	0.06	0.79 (−0.07 to 1.66)	0.07
Marital status	2.94 (1.77 to 4.12)	<0.001			2.36 (1.15 to 3.58)	<0.001	2.36 (1.13 to 3.58)	<0.001
Household Number	0.27 (−0.00 to 0.54)	0.05			0.32 (0.07 to 0.57)	<0.05	0.31 (0.06 to 0.56)	<0.05
Mealtime with Family	0.74 (0.37 to 1.11)	<0.001			0.46 (0.09 to 0.83)	<0.05	0.48 (0.11 to 0.86)	<0.05
Alcohol consumption	−0.59 (−1.41 to 0.23)	0.16					0.26 (−0.78 to 1.31)	0.62
Family history of diabetes	−0.03 (−0.05 to −0.00)	0.03					−0.01 (−0.03 to 0.01)	0.30

Note: Univariate represents univariable robust regression, and models represents multivariable robust regression analysis model. ^#^ Treated as categorical data, *p* value < 0.05 statistically significant, *p* value < 0.001 statistically strongly significant. Powers of model II and model III were 80% with the sample of 300. Model I: adjustment for age, family income, educational attainment, model II: adjustment for sex, age, family income, educational attainment, marital status, household number, mealtime with family, model III: adjustment for sex, age, family income, educational attainment, marital status, household number, mealtime with family, alcohol consumption, family history of diabetes. Abbreviation: CI, Confidence Interval.

## Data Availability

The data analyzed in this study will be available on request from the corresponding author as there are ongoing analysis.

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
