# Peer review of "Social Support between Diabetes Patients and Non-Diabetes Persons in Yangon, Myanmar: A Study Applying ENRICHD Social Support Instrument"

_ijerph, 2021, doi:10.3390/ijerph18147302_

Round 1

Reviewer 1 Report

A prospective study examining the need for social support of people with diabetes and the control group. The research tool was a standardized questionnaire.

Doubts are raised by the very wide age range of the studied group and its small size, which may make general inference difficult.

The developed research results may be an important aspect in undertaking actions for public health or care for the patient and his family.

Author Response

We revised the manuscript addressing each comment of reviewer 1.

Point 1: A prospective study examining the need for social support of people with diabetes and the control group. The research tool was a standardized questionnaire.

Response 1: Thank you very much for the advice. We added the recommendation in the discussion. Line 449-468.

Point 2: Doubts are raised by the very wide age range of the studied group and its small size, which may make general inference difficult.

Response 2:  We calculated the power of regression analysis, applying sample power calculation in STATA. Power of the Model 2 with 8 covariates and Model 3 with 10 variates were 80% with the sample of 300.

We have added under the table4, line 287-288.

Amendment: “Power of the Model 2 was, and Model 3 were 80% with the sample of 300.”

Point 3: The developed research results may be an important aspect in undertaking actions for public health or care for the patient and his family.

Response 3: We highlighted the point as reviewer advised: Line 444-446

Reviewer 2 Report

see attachment

Author Response

Thank you very much for your insightful comments on revising the abstract and other aspects of the paper.

We would like to give response for your comments as following.

Point 1: Title: “appling” should read “applying”

Response 1: We corrected “Applying” in the title.

Point 2: Abstract –

Overall, it is ok. Perhaps consider putting p values for the significant relationships seen between social support and various independent variables.

Response 2: We added the relevant p values in the line 30.

Point 3: Line 77: consider elaborating how implementation of low-cost peer support was cost effective: less visits to hospitals, greater compliance to medications, etc.

Response 3: We reviewed and edited according to the comment in line 79-80.

Point 4: Line 86: add the word “with” in sentence “how they interact each other on diabetes...”

Response 4: We edited in line 89.

Point 5: Specific hypothesis in this case-control study is not provided.

Response 5: We hypothesize that there will be difference between social support of cases (diabetes patients) and control (people without diabetes) and revised in line 110-112.

Point 6: Material and Methods –

Line 165: “Centra obesity” should read “central obesity.”

Response 6: We edited in line 177.

Point 7: The exclusion criteria states anyone with a serious illness. What constituted serious illness, and did they exclude those with psychiatric illnesses too? Also, were pregnant women also included?

Response 7: We added the example of serious illness and excluded pregnant women in the study. Line 107-109

Point 8: Rephrase sentence 178-180; how were questionnaires given to participants.

Response 8: The printed questionnaires were distributed, and the participants were interviewed by a research assistant during the data collection, line 191 – 201.

Point 9: Results –

Table 4: was there a reason BMI was not included as a covariate?

Response 9: In the univariate analysis, BMI is not statistically significant, and we supposed that it may be not directly related to the social support. So, we left BMI in the multivariate analysis.

Point 10: In footnote of all tables, perhaps add the p value considered statistically significant.

 Response 10: We added the significant p values in the footnote of all tables (table 1 to 4).

Point 11: Discussion

The discussion needs significant revisions. Specifically, why are the findings of this study important in this population specifically, when compared to other ethnic groups? The introduction does a great job discussing why social support in diabetic patients is important and would be helpful to discuss and expand on this in the discussion.

Response 11: We discussed the role of social support and family in diabetes patients; line 436-446.

Point 12: Lines 296-317: Authors discuss a previous study that suggests several factors that may contribute to diabetes in the urban areas of Yangon. It would be interesting if there are any studies that looked at the role of culture and health outcomes, including perceived social support, especially in the continent of Asia.

Response 12: We reviewed a study in Myanmar and discussed for the cultural and social behaviours in line 336-341.

Point 13: Line 321: “among the diabetes” should say “among the diabetes group”

Response 13: We added “group” in line 357.

Point 14: Line 333: rephrase 331 “perhaps these might be impact of diabetes-related health education.” Also, it would be interesting as to why this study found no differences in social support between two genders.

Response 14: We found no statistical significance between social support (both each item of ESSI and total perceive social support) among male and female participants. So, we are hesitant to discuss or highlight this point in our discussion.

Point 15: Line 339: “patient” should say ‘patients”

Response 15: We corrected “patients” in line 377.

Point 16: Paragraph lines 334-335: It may be noteworthy to mention no validated measures of stress (anxiety, psychosocial stress, acculturative stress) in the current study, which may explain some of the significant associations.

Response 16: Thank you so much for your insightful comment. We would like to measure the disease-related distress and care burden among same population in the future, but we will mainly discuss on social support which examined in this study.

Reviewer 3 Report

This is a clearly written and interesting paper investigating levels of perceived social support among individuals with and without diabetes in Yangon, Myanmar.  The fact that this study was conducted in an understudied region is important.  The findings are interesting, and clearly communicated.  I have a few comments, which should be addressed before being accepted.

  1. In the Methods section, lines 154-156, "ethnics" should be changed to "ethnicity", and please indicate and describe how each of the variables were measured/coded.  It describes education and some additional variables below, but not all variables were described.
  2. Line 161, recommend changing the word "habit" to "behavior" throughout the manuscript.  Readers in the US will detect a negative connotation to the word "habit", which should be avoided.
  3. Line 165, please specify how blood pressure was measured, with auscultation or oscillometric.
  4. In the conclusion, line 431, recommend changing the word "normal" to "healthy", or just saying "people without diabetes."

Author Response

Thank you very much for your insightful comments. We revised the manuscript addressing each comment of reviewer 3.

Point 1: In the Methods section, lines 154-156, "ethnics" should be changed to "ethnicity", and please indicate and describe how each of the variables were measured/coded.  It describes education and some additional variables below, but not all variables were described.

Response 1: We corrected "ethnicity" and described how other variables were measured in line 162-172. 

Point 2: Line 161, recommend changing the word "habit" to "behavior" throughout the manuscript.  Readers in the US will detect a negative connotation to the word "habit", which should be avoided.

Response 2: We edited as "smoking behavior" in line 172. 

Point 3: Line 165, please specify how blood pressure was measured, with auscultation or oscillometric.

Response 3: The blood pressure was measured with automatic oscillomertic method and we edited in line 179-183.

Point 4: In the conclusion, line 431, recommend changing the word "normal" to "healthy", or just saying "people without diabetes."

Response 4: It has changed to "people without diabetes" in line 486.
